# Sensor Based on a Solid Oxide Electrolyte for Measuring the Water-Vapor and Hydrogen Content in Air

Anatoly Kalyakin [1], Anatoly K. Demin [1,2,*], Elena Gorbova [1,2,3], Alexander Volkov [1] and Panagiotis E. Tsiakaras [1,2,3,*]

1   Laboratory of Electrochemical Devices Based on Solid Oxide Proton Electrolytes, Institute of High Temperature Electrochemistry, Russian Academy of Sciences, 20 Akademicheskaya Str., 620990 Yekaterinburg, Russia
2   Institute of Chemical Engineering, Ural Federal University, 19 Mira Str., 620002 Yekaterinburg, Russia
3   Laboratory of Alternative Energy Conversion Systems, Department of Mechanical Engineering, School of Engineering, University of Thessaly, 1 Sekeri Str., Pedion Areos, 38834 Volos, Greece
*   Correspondence: akdemin004@rambler.ru (A.K.D.); tsiak@uth.gr (P.E.T.)

**Abstract:** The present communication describes the results of the performance and the assessment of a sensor based on a solid oxide electrolyte with a composition of $0.9ZrO_2 + 0.1Y_2O_3$ (YSZ), equipped with a ceramic diffusion barrier for measuring the water vapor and hydrogen content in air. The possibility of determining the concentration of water vapor and hydrogen in the air is based on the measurement of the limiting current value. For the calculation of the steam and hydrogen concentration in ambient air, analytical expressions were obtained and applied, using the limiting current values measured in air with a standard oxygen concentration of 20.9 vol.% and in the analyzed air. A two-stage method for the determination of the hydrogen and steam amount in ambient air is proposed. It is stated that the sensor operates successfully at the temperature of 700 °C and can be applied for the continuous determination of steam or hydrogen concentrations in air.

**Keywords:** electrochemical cell; amperometric sensor; hydrogen and steam detection; diffusion barrier; limiting current; solid oxide electrolyte





## 1. Introduction

Currently, zirconia-based solid electrolytes with oxygen ion conductivity are widely used in various electrochemical devices, such as fuel cells, oxygen pumps, and electrolyzers. The widest application of solid electrolytes is in gas sensors operating at elevated temperatures. Potentiometric and amperometric solid electrolyte sensors for oxygen control are widely used in energetics, metallurgy and transport [1–5]. The designs of different types of oxygen sensors and their utilized materials are described in previously published works [5–10], while the investigations of sensors for humidity determination based on solid electrolytes with oxygen ion [11,12] and proton [13–17] conductivity, as well sensors based on a combination of both types of solid oxide electrolytes [18] have been performed successfully. For instance, the humidity value was determined by analyzing a ratio of the second limiting current to the first one. Hydrogen sensors based on various proton-conducting electrolytes, namely, Nafion for measuring the hydrogen concentration in a $N_2 + H_2$ mixture [19], Nasicon for hydrogen detection in oxidizing and non-oxidizing atmospheres [20], as well as a potentiometric sensor based on a YSZ oxygen ion conducting electrolyte for measuring the low (0 to 0.145%) hydrogen concentration in air [21], were also studied.

To the best of the authors' knowledge, no attempts have been made to use amperometric sensors based on an oxygen-ion electrolyte to determine the hydrogen in air. Therefore, the developed sensor has a certain novelty in this respect. In the present work is an amperometric sensor based on a $0.9ZrO_2 + 0.1Y_2O_3$ electrolyte, which possesses oxygen-ion conductivity in a wide range of temperatures and oxygen concentrations in an ambient media.

An important part of the amperometric sensor is a diffusion barrier served for a gas diffusion from the ambient atmosphere into the sensor's inner chamber. As a rule, a porous layer or a laser-drilled hole are used as the diffusion barrier [8]. In some recent papers, a dense layer made of materials possessing a mixed ionic-electronic conductivity was used as the diffusion barrier [9,22,23]; however, it is difficult to control the parameters of the above-mentioned diffusion barriers and even more difficult to reproduce them. At the same time, the sensor characteristics, namely, a limiting current, depends on the diffusion barrier parameters.

In the present study, a thin wall ceramic capillary was used as the diffusion barrier. The advantage of a ceramic capillary compared to a metal capillary is the resistance to oxidation during hermitization occurring in an oxidizing atmosphere, which can cause a decrease in the internal diameter of the metallic capillary. The advantage of a ceramic capillary compared to porous media has an easy reproducibility and repeatability.

*Background of the Amperometric Sensor Operation*

The amperometric sensor based on an oxygen ion electrolyte works under a regime of oxygen pumping out from the inner chamber. It is schematically represented in Figure 1.

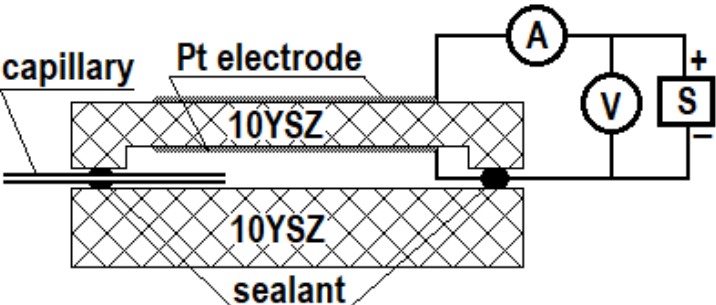

**Figure 1.** Schematic representation of an amperometric sensor based on an oxygen ion electrolyte equipped with a capillary as a diffusion barrier.

When a positive potential is applied to the external electrode of the sensor, oxygen is pumped out of the sensor chamber due to the transfer of oxygen ions through the electrolyte to this electrode. The corresponding reaction at the external electrode is as follows:

$$O^{-2} = 0.5O_2 + 2e^- \tag{1}$$

and the reverse reaction occurs at the inner electrode. The sensor current increases with the applied voltage increase and the limiting current appears at a definite voltage. The limiting current is determined by the oxygen concentration in the analyzed gas and its value obeys the following equation:

$$I_{lim} = -\frac{4FD_{O_2}SP}{RTL}\ln\left(1 - X_{O_2}\right) \tag{2}$$

where F is the Faraday constant, $D_{O_2}$ is the diffusion coefficient of oxygen in the gas mixture, S is the cross-section area of the diffusion barrier, P is the absolute gas pressure, R is the gas constant, T is the temperature in Kelvin, L is the length of the diffusion channel, and $X_{O_2}$ is the oxygen mole fraction. When the oxygen concentration around $X_{O_2} \approx 0.2$ changes in a narrow range ($\pm 0.02$), it is acceptable to use the simpler linear equation:

$$I_{lim} = k\frac{4FD_{O_2}SP}{RTL}X_{O_2} \tag{3}$$

where k = 1.12 is the coefficient taking into account the logarithmic dependence of the limiting current on the oxygen concentration. Note that this k value is valid only for the mentioned $X_{O_2}$ value and for other $X_{O_2}$ values the coefficient k is different.

The oxygen concentration in dry air, expressed as a mole fraction of oxygen, is $X_{O_2}^0 = 0.209$. Any impurity, in particular water vapor, causes a decrease in the concentration of oxygen in the air. The relationship between the mole fraction of oxygen in the humid air, $X_{O_2}'$, and the humidity of the air, expressed in the mole fraction of steam, is as follows:

$$X_{O_2}' = X_{O_2}^0 \cdot \left(1 - X_{H_2O}\right) \tag{4}$$

If the analyzed air contains hydrogen, then the latter interacts with oxygen at the operating temperature of the sensor, thus, reducing the oxygen concentration. Here we consider the case when hydrogen is an admixture to dry air. The mole fraction of oxygen in the air after hydrogen combustion, $X_{O_2}''$, (below called 'hot air') can be calculated using the following equation:

$$X_{O_2}'' = \frac{X_{O_2}^0 \cdot \left(1 - X_{H_2}\right) - 0.5 \cdot X_{H_2}}{1 - 0.5 \cdot X_{H_2}} \tag{5}$$

With great accuracy, this equation can be simplified to the following:

$$X_{O_2}'' = X_{O_2}^0 \cdot \left(1 - 0.5 \cdot X_{H_2}\right) - 0.5 \cdot X_{H_2} \tag{6}$$

Equations (4)–(6) clearly show that the concentration of oxygen in hot air decreases as the concentration of steam or hydrogen increases; therefore, the limiting current of the amperometric sensor depends on the content of these impurities in the air. In order to use the amperometric sensor for the determination of steam or hydrogen concentration in air, it is possible to experimentally measure the dependences of the limiting currents on the steam concentration in air and on the hydrogen concentration in dry air. Having these graphical calibration dependences, it is possible to determine the content of these impurities from the obtained value of the limiting current in the analyzed gas mixture.

On the other hand, the desired concentrations can be calculated analytically. If the limiting currents $I_{lim}\left(X_{O_2}^0\right)$ and $I_{lim}\left(X_{O_2}\right)$ in dry and humid air, respectively, are known, the oxygen mole fraction in the second case can be found using the following equation:

$$X_{O_2} = X_{O_2}^0 \cdot \frac{I_{lim}\left(X_{O_2}\right)}{I_{lim}\left(X_{O_2}^0\right)} \tag{7}$$

The mole fraction of steam in humid air can be calculated from Equation (4):

$$X_{H_2O} = \frac{X_{O_2}^0 - X_{O_2}'}{X_{O_2}^0} \tag{8}$$

The mole fraction of hydrogen in dry air can be calculated from the Equation (6):

$$X_{H_2} = 2 \cdot \frac{X_{O_2}^0 - X_{O_2}''}{1 + X_{O_2}^0} \tag{9}$$

Substituting $X_{O_2}$ from Equation (7), instead of in Equation (8) in the case of the humid air analysis and the same instead of $X_{O_2}''$ in Equation (9) in the case of the air–hydrogen mixture analysis, it is possible to calculate the content of the corresponding admixture in air. On the other hand, Equation (7) can be used for the calculation of the limiting current depending on the admixture concentration in air:

$$I_{lim}\left(X_{O_2}\right) = I_{lim}\left(X_{O_2}^0\right) \cdot \frac{X_{O_2}}{X_{O_2}^0} \tag{10}$$

Substituting $X'_{O_2}$ from Equation (4) and $X''_{O_2}$ from Equation (6) instead of $X_{O_2}$ in Equation (10), one can obtain the theoretical dependence of the limiting current on the concentration of steam and hydrogen in air, respectively.

It seems that the presence of steam the in air–hydrogen mixture does not allow for determining the hydrogen concentration due to an additional decrease in the oxygen concentration compared to the dry air–hydrogen mixture; therefore, it is necessary for the air to be dried before analysis with the amperometric sensor. Zeolites of phosphorous oxide $P_2O_5$ can be used for this purpose.

However, it is possible to use a two-stage procedure to measure the content of both admixtures in atmospheric air. The first stage involves measuring the limiting current in atmospheric air. The second stage involves measuring the limiting current in dried atmospheric air. If the limiting currents are the same, this indicates that the atmospheric air is free of hydrogen. Otherwise, the first limiting current is lower than the second one, signaling the presence of hydrogen in the air. Using these limiting currents, the oxygen concentrations in hot atmospheric air containing hydrogen $\left(X'''_{O_2}\right)$ and in dried atmospheric hot air $\left(X''_{O_2}\right)$ can be derived from Equation (7). The hydrogen concentration is calculated according to Equation (9). The steam concentration can be calculated according to the following equation:

$$X_{H_2O} = \frac{X''_{O_2} - X'''_{O_2}}{X''_{O_2}} \tag{11}$$

Thus, both the steam and hydrogen concentrations in atmospheric air can be measured by the same sensor using the two-stage method described above.

Moreover, it is possible to calculate the hydrogen mole fraction in humid air if the air humidity is known, for instance, as measured by a moisture meter. The relationship between the mole fraction of oxygen and the mole fractions of the steam and hydrogen in this case can be found using the following equation, which combines Equations (4) and (6):

$$X'''_{O_2} = \left(X^0_{O_2}\left(1 - 0.5 \cdot X_{H_2}\right) - 0.5 \cdot X_{H_2}\right) \cdot \left(1 - X_{H_2O}\right) \tag{12}$$

Then, the following equation can be derived to calculate the mole fraction of hydrogen:

$$X_{H_2} = 2 \cdot \frac{X^0_{O_2} - X'''_{O_2} - X^0_{O_2} \cdot X_{H_2O}}{1 + X^0_{O_2} - 2 \cdot X^0_{O_2} \cdot X_{H_2O} - X_{H_2O}} \tag{13}$$

In summary, it is possible to calculate the hydrogen content in a wet mixture of $H_2$ + air if the mixture's humidity is known and the oxygen mole fraction in the hot mixture is obtained from Equation (7).

## 2. Results and Discussion

At the first stage, dry air was supplied into the furnace interior and the current-voltage dependence of the sensor in dry air was measured. The fluctuations of the measured sensor current at a fixed applied voltage did not exceed 10 µA. The time of the current measurement at a fixed voltage was 1 min, and the interval between measurements was 10 s. The obtained current values were averaged.

It is important to note that the obtained value of the limiting current of the sensor in dry air was used to calculate the dependences of the limiting current on the concentration of hydrogen in the dry air and on the humidity of the air in accordance with Equation (10). At the next stage, mixtures of dry air with hydrogen of various concentrations were supplied to the furnace interior and the current-voltage dependences for each composition were measured.

Figure 2a shows the volt-ampere dependences of the sensor measured in 'dry air'–hydrogen mixtures, containing hydrogen of various concentrations. As seen, the limiting

current plateaus begin to form at the voltage value of 0.4 V. It is clear that the limiting currents change insignificantly with the hydrogen concentration change. The difference in the volt-ampere dependences becomes more noticeable in Figure 2b, where a narrow section of the Y axis is selected.

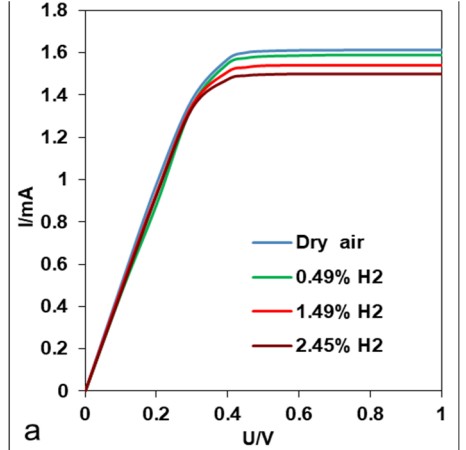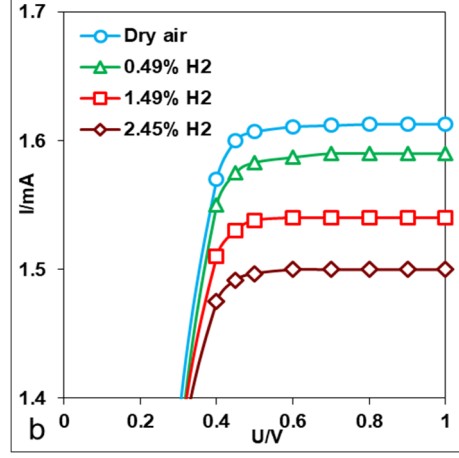

**Figure 2.** Volt-ampere dependences of the sensor in a 'dry air'–hydrogen mixture at different hydrogen concentrations. The full Y-axis (**a**) and the selected scale of the Y-axis (**b**).

The analogical measurements were performed when air with different humidity amounts was supplied to the furnace interior. Figure 3a shows the volt-ampere characteristic curves of the sensor, measured in humidified air containing various concentrations of steam. As in the previous case, the limiting currents change insignificantly as the steam concentration changes. Obviously, precise measurements of the limiting currents are necessary in both cases.

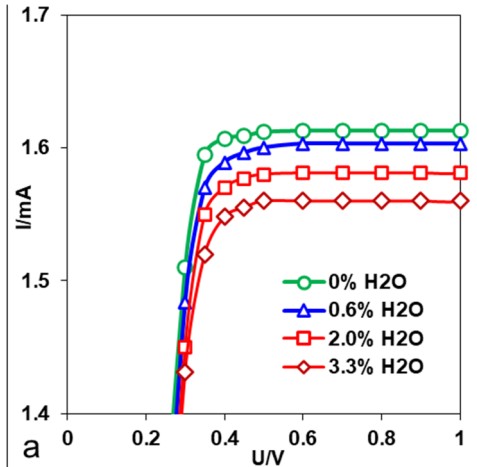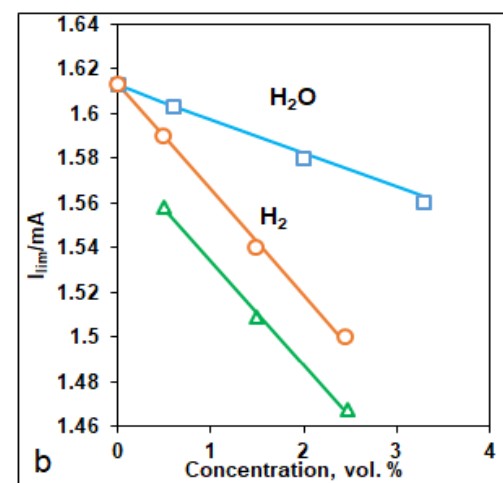

**Figure 3.** Volt-ampere dependences of the sensor in air with different humidity (**a**); and dependences of the limiting currents on the steam concentration in air (rectangles), hydrogen concentration in dry air (circles) and hydrogen concentration in humid (3.3%) air (triangles) (**b**). Symbols are the experimental data; straight lines are theoretically calculated according to Equation (10).

Figure 3b shows the limiting current dependences on the steam (squares) and hydrogen concentration (circles) in the air. It is seen that the limiting current dependence on the steam concentration is weaker (lower slope) than that on the hydrogen concentration. Moreover, as can be seen, at the same time, the experimental data of both dependences are very close to the theoretically-calculated lines according to Equation (10).

The measurement of the limiting current dependence in the case of the humidified $H_2$ + air mixture was performed. Gas mixtures of $H_2$ + air with different hydrogen concentrations were passed through the bubbler at 26 °C to provide the mixture humidity of 3.3%. The limiting current dependence on the hydrogen concentration in the humid air is shown in Figure 3b with the green triangles. Obviously, this dependence is parallel to the dependence of the limiting current obtained in the dry air, and lower than the latter by the difference between the limiting currents in the dry and humid air. Therefore, it is possible to use the calibrated dependences obtained in humid air and a $H_2$ + dry air mixture for the determination of the hydrogen concentration in humid air for which the humidity is known. To do this, one can build a line that lies below the calibrated dependence obtained in a dry mixture of $H_2$ + air, by the difference in the limiting currents in dry and humid air with a known humidity. Then, using the as-built calibrated dependence, the hydrogen concentration corresponding to the measured limiting current can be found.

## 3. Experimental Procedure

The sensor shown in Figure 4 is an amperometric one and it consisted of two YSZ plates. The low rectangular plate is 16 mm long, 10 mm wide and 1 mm thick, while the upper oval plate has slightly less size and the same thickness.

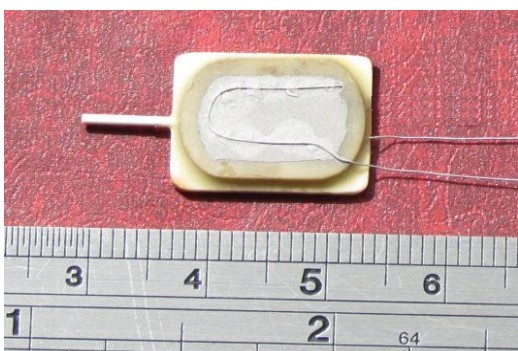

**Figure 4.** Photo of the as-fabricated amperometric sensor equipped with a capillary as a diffusion barrier. A ruler allows for estimating the dimensions of the sensor.

The plates were cut from the electrolyte plates produced by the Chepetsk mechanical plant (Glazov, Russia). As seen, the upper plate has a recess of 9 mm in length, 6 mm in width and 0.5 mm in depth made by a diamond tool. Pt porous electrodes were placed at the opposite sides of this plate, while Pt wires of 0.1 mm in diameter were used as the current leads. The plates were sealed by a high temperature glass sealant. The ceramic capillary was hermetically sealed between the plates. The inner channel diameter of the capillary was $350 \pm 5$ microns, and its length was $11.0 \pm 0.1$ mm. The sensor was placed in a tube furnace with an internal volume of 0.1 L. The temperature in the furnace was maintained at 700 °C with an accuracy of $\pm 2°$, by the aid of a thermoregulator TP 703 "Varta".

The gas set-up is schematically shown in Figure 5. Cylinders with compressed dry air and calibration gas mixtures of 'air + hydrogen' with a hydrogen concentration of 0.49, 1.49 and 2.45 vol.% (LLC "PGS-service", Sverdlovsk region) were used in the experiments.

The gas mixture flow rate of 20 mL/min was set by a mass-flow controller (F-201C-33-V, Bronkhorst). The air–hydrogen mixture was fed directly into the interior of the furnace. The air was passed through a humidifier (bubbler) at 0, 18 and 26 °C and saturated with 0.6, 2.0 and 3.3% humidity, respectively. The humidity content was controlled with an IVTM-7N-17 sorption-capacitive moisture meter with a relative error of 4%. A GPS-18,500 current source was used to supply voltage to the sensor. The voltage was controlled with a GDM-8246 multimeter with an accuracy of 10 mV. The sensor current was measured with a multimeter of the same type with an accuracy of 1 µA.

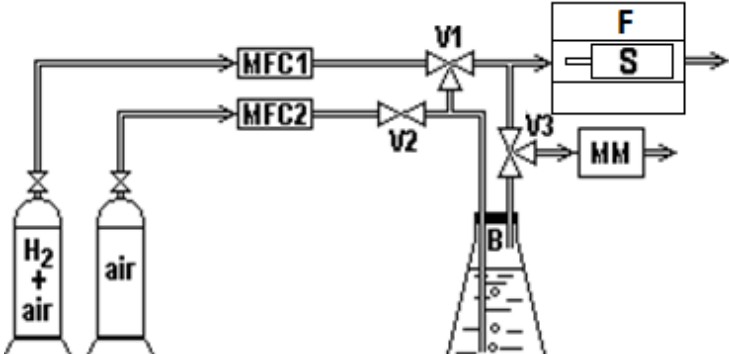

**Figure 5.** Mass-flow diagram. MFC are mass-flow controllers; V1 and V3 are three-way valves; V2 is a two-way valve; F is a furnace; S is a sensor; MM is a moisture meter; B is a bubbler.

## 4. Conclusions

For the first time, it is demonstrated that the amperometric sensor based on a YSZ electrolyte equipped with a ceramic capillary as a diffusion barrier can be utilized for the determination of the steam and/or hydrogen concentration in air. For the determination of the hydrogen concentration in ambient air, preliminarily it is necessary for the air to be dried. The latter procedure can be carried out by passing the air through a dryer filled with zeolite or phosphorus pentoxide $P_2O_5$. Analytical expressions for calculation of the steam and hydrogen concentration in ambient air using the limiting current values in air with the standard oxygen concentration of 20.9 vol.% and in the analyzed air were obtained. The two-stage procedure for the determination of the concentrations of hydrogen and steam in ambient air was also described. Because of a rather weak dependence of the limiting current on the hydrogen and especially the steam concentration in air, the limiting current values need to be measured with a high accuracy. The developed sensor can be applied for a control of the ambient air where a hydrogen appearance is possible, for instance, near hydrogen fuel cells, near hydrogen tanks, near devices for hydrogen production, etc. It should be noted that the presence of any combustible substances leads to a decrease in the oxygen content under the working condition of the sensor; therefore, the developed sensor can be used if the appearance of any impurities other than steam and hydrogen is impossible or where their concentrations are very low.

**Author Contributions:** Investigation, Data Curation, Validation, Writing—Original Draft, Methodology, A.K., E.G. and A.V.; Supervision, Project Administration, Resources, A.K.D.; Conceptualization, Writing—Review and Editing, P.E.T. All authors have read and agreed to the published version of the manuscript.

**Funding:** This research received no external funding.

**Data Availability Statement:** The data that support the findings of this study are available from the corresponding authors upon reasonable request.

**Conflicts of Interest:** The authors declare no conflict of interest.

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
