# Peer review of "Sensor Based on a Solid Oxide Electrolyte for Measuring the Water-Vapor and Hydrogen Content in Air"

_catalysts, doi:10.3390/catal12121558_

Round 1

Reviewer 1 Report

Manuscript entitled "Sensor based on solid oxide electrolyte ......" reports on YSZ electrolyte based sensor for  steam and/or hydrogen concentration in air. The two- 18 stage method for the determination of hydrogen and steam amount in ambient air is proposed. Results supprots their claim. Manuscript is well written and the results are interesting. Manuscript can be accepted in the present format. 

Author Response

REVIEWER #1

Manuscript entitled "Sensor based on solid oxide electrolyte ......" reports on YSZ electrolyte based sensor for steam and/or hydrogen concentration in air. The two- 18 stage method for the determination of hydrogen and steam amount in ambient air is proposed. Results supports their claim. Manuscript is well written and the results are interesting. Manuscript can be accepted in the present format. 

We thank you very much for the reviewer’s decision.

Reviewer 2 Report

The authors presented an interesting work that demonstrated an amperometric sensor based on a YSZ-214 electrolyte with a ceramic capillary as a diffusion barrier could be used to determine the concentration of vapor and / or hydrogen in the air. However, I do have some suggestions and comments that I think will help increase the impact of this manuscript. A small modification is needed before it is accepted for publication.

1.        The advantages of thin-walled ceramic capillary tubes over other diffusion barriers are not reflected.

2.        Figure 4 a and figure 4 b are the same set of data. It is too cumbersome to divide them into two figures.

3.  There are several format errors in this article. For example, the text portion of Figure 5a.

4.  The significance and innovation of this work should be highlighted,such as in the Introduction or Conclusion section.

Author Response

REVIEWER #2

  1. The advantages of thin-walled ceramic capillary tubes over other diffusion barriers are not reflected.

Thank you for this advice. We added to the text the following sentences: “The advantage of a ceramic capillary compared to a metal capillary is the resistance to oxidation during hermitization occurring in an oxidizing atmosphere, which can cause a decrease in the internal diameter of the capillary. The advantage of a ceramic capillary compared to porous media is easy reproducibility and repeatability”.

  1. Figure 4a and figure 4b are the same set of data. It is too cumbersome to divide them into two figures.

Thank you for this remark. However, we consider that it is reasonable to show very little differences in the limiting currents when the full Y-axis is used (Figure 4a) and more distinct differences in the limiting currents when a selected part of Y-axis is used (Figure 4b).

  1. There are several format errors in this article. For example, the text portion of Figure 5a.

We have corrected the text portion at Figure 5a.

  1. The significance and innovation of this work should be highlighted, such as in the Introduction or Conclusion section.

Following the reviewer’s suggestion, in the Introduction section we wrote: “To the best of the authors' knowledge, no attempts have been made to use amperometric sensors based on an oxygen-ion electrolyte to determine hydrogen in air”. We added the following: “Therefore, the developed sensor has a certain novelty in this respect”.  At the beginning of the Conclusion section, it is written: “For the first time, it is demonstrated that the amperometric sensor based on YSZ electrolyte equipped with a ceramic capillary as a diffusion barrier can be utilized for the determination of steam and/or hydrogen concentration in air”.

 Moreover, in the Conclusion section we reported: “The developed sensor can be applied for a control of ambient air where hydrogen appearance is possible”. We added the following: “For instance, near hydrogen fuel cells, near hydrogen tanks, near devices for hydrogen production, etc.”

Reviewer 3 Report

In this study, the possibility of determining the concentration of water vapor and hydrogen in the air is based on the measurement of the limiting current value. Analytical expressions were obtained and applied for the calculation of steam and hydrogen concentration in ambient air, using the limiting current values measured in air with standard oxygen concentration of 20.9 vol. % and in analyzed air. The two-stage method for the determination of hydrogen and steam amount in ambient air is proposed. It is stated that the sensor operates successfully at the temperature of 700 ℃ and it can be applied for the continuous determination of steam or hydrogen concentration in air. However, some problems should be revised before its publication: 

1. Please pay more attention on the language. For example, in abstract section, the sentences are not concise enough.

2. The authors only tested data where the furnace temperature was held at 700°C and whether the performance of this amperometric sensor at other temperatures was also tested.

3. The author talks about “The air-hydrogen mixture was fed directly into the interior of the furnace.” The interior of the furnace is not visible in figure 3.

4.Please note the correct writing format of some symbols.

Author Response

REVIEWER #3

  1. Please pay more attention on the language. For example, in abstract section, the sentences are not concise enough.

Answer: We have carefully checked the entire manuscript and made the appropriate improvement.

  1. The authors only tested data where the furnace temperature was held at 700°C and whether the performance of this amperometric sensor at other temperatures was also tested.

Thank you for this question. In fact, we chose a temperature of 700 ° C, since at lower temperatures the current-voltage dependencies were not ideal (deviation from a straight line at the beginning of the I-V dependences)

  1. The author talks about “The air-hydrogen mixture was fed directly into the interior of the furnace.” The interior of the furnace is not visible in figure 3.

Thank you for this question. We corrected Figure 3 and added a furnace with a sensor inside.

  1. Please note the correct writing format of some symbols.

Honestly, we don’t understand what the reviewer means. Could she/he please give us an example?